# High Incidence of Candidemia in Critically Ill COVID-19 Patients Supported by Veno-Venous Extracorporeal Membrane Oxygenation: A Retrospective Study

**DOI:** 10.3390/jof9010119

**Published:** 2023-01-14

**Authors:** Francesco Alessandri, Giancarlo Ceccarelli, Giuseppe Migliara, Valentina Baccolini, Alessandro Russo, Carolina Marzuillo, Mariateresa Ceparano, Giovanni Giordano, Pierfrancesco Tozzi, Gioacchini Galardo, Giammarco Raponi, Claudio Mastroianni, Mario Venditti, Francesco Pugliese, Gabriella d’Ettorre

**Affiliations:** 1Intensive Care Unit, Department of General, Specialistic Surgery Sapienza University, 00185 Rome, Italy; 2Azienda Ospedaliero Universitaria Policlinico Umberto I, 00161 Rome, Italy; 3Department of Public Health and Infectious Diseases, Sapienza University, 00185 Rome, Italy; 4Unit of Infectious and Tropical Diseases, Department of Medical and Surgical Sciences, “Magna Graecia” University, 88100 Catanzaro, Italy

**Keywords:** COVID-19, candida, mycosis, veno-venous ECMO

## Abstract

Background: The incidence of candidemia in severe COVID-19 patients (0.8–14%) is two- to ten-fold higher than in non-COVID-19 patients. Methods: This retrospective analysis aimed to analyse the incidence of bloodstream infections (BSI) due to Candida in a cohort of COVID-19 patients supported with ECMO. Results: Among 138 intubated and ventilated patients hospitalized for ≥10 days in the intensive care unit of a teaching hospital, 45 (32.6%) patients received ECMO support, while 93 patients (67.4%) did not meet ECMO criteria and were considered the control group. In the ECMO group, 16 episodes of candidaemia were observed, while only 13 in patients of the control group (36.0% vs. 14.0%, *p*-value 0.004). It was confirmed at the survival analysis (SHR: 2.86, 95% CI: 1.39–5.88) and at the multivariable analyses (aSHR: 3.91, 95% CI: 1.73–8.86). A higher candida score seemed to increase the hazard for candidemia occurrence (aSHR: 3.04, 95% CI: 2.09–4.42), while vasopressor therapy was negatively associated with the outcome (aSHR: 0.15, 95% CI: 0.05–0.43). Conclusions: This study confirms that the incidence of candidemia was significantly higher in critically ill COVID-19 patients supported with VV-ECMO than in critically ill COVID patients who did not meet criteria for VV-ECMO.

## 1. Introduction

Severe acute respiratory syndrome-coronavirus 2 (SARS-CoV-2) infection ranges from asymptomatic to life-threatening conditions; in severe cases, respiratory failure may require support with invasive mechanical ventilation (IMV) and extracorporeal membrane oxygenation (ECMO) in intensive care units (ICU) [1]. Among coronavirus infectious disease 2019 (COVID-19) critically ill patients, superinfections can contribute to a more severe clinical course and longer hospital stay [2]. Previous studies demonstrated that the incidence of candidemia in severe COVID-19 patients (0.8–14%) is two- to ten-fold higher than in non-COVID-19 patients [3,4,5]. During this pandemic, the increase in the number of patients admitted to ICU made this worse [6]. SARS-CoV-2, effect on the digestive tract and a deficiency in the interferon pathway anti-*Candida* immune response have been investigated [7,8]. Prolonged length of ICU stay is a risk factor for developing fungal superinfections, especially invasive candidiasis due to many factors including use of broad-spectrum antibiotics or corticosteroids, invasive procedures, and multiple devices such as central venous lines, intubation tubes, continuous renal replacement therapy (CRRT) and ECMO [9,10,11]. Currently, few pieces data are available on infectious risks in patients with COVID-19 supported with ECMO; however, this extracorporeal support was previously associated with an increased risk of bloodstream infections (BSI) in non-COVID-19 population. In particular, in large pre-pandemic studies, *Candida* accounted for 5.4–8.9% of all BSIs [12,13].

To the best of our knowledge, there are not studies that analyse the incidence of *Candida* species bloodstream infections in patients affected by COVID-19 undergoing veno-venous (VV)-ECMO. 

Therefore, we designed this study to analyse the incidence of BSI due to Candida in a cohort of COVID-19 patients supported with ECMO and hospitalized in the intensive care of a large ECMO reference centre in Central Italy.

## 2. Materials and Methods

### 2.1. Design of the Study, Population, Settings, Data Collection and Outcomes

This study is a “real life” retrospective analysis of all adults (>18 years) suffering from bilateral interstitial pneumonia due to SARS-CoV-2 infection, supported with IMV or IMV plus ECMO admitted between March 2020 and June 2021 to the ICU of Policlinico Umberto I, University “Sapienza”, Rome, Italy for ≥10 days. We screened all patients for ECMO, following the standard ELSO guidelines criteria: severe ARDS with hypoxia (PaO2:FIO2 < 100; PaO2:FIO2 < 150 with concern for progressive/quick clinical decline) and/or hypercapnia (PaCO2 > 60 mmHg) despite the use of advanced ventilator settings and adjunctive therapies (i.e., prone positioning, paralysis) [14]. We considered exclusion criteria for ECMO: mechanical ventilation >10 days, significant underlying co-morbidities (CKD: ≥III cirrhosis, irreversible neurologic injury, potential disseminated malignancy) and other pre-existing life-limiting medical conditions, severe multiple organ failure, uncontrolled bleeding and contraindications to anticoagulation. 

All patients clinically diagnosed with COVID-19 underwent a high-resolution CT/non-contrast enhanced chest CT at admission in the hospital. 

The sources for patient data were medical records stored in the electronic information system of the ICU involved. The variables considered included: past clinical history (comorbidities), current clinical history, treatment, ventilation parameters and laboratory data. The study’s primary endpoint was the assessment of rate and time to in-hospital all-cause mortality. Secondary endpoint was the incidence of different *Candida* species as the cause of candidemia.

### 2.2. Diagnosis of SARS-CoV-2 Infection

The COVID-19 case definitions of European Centre for Disease Prevention and Control (ECDC) was adopted [15]. A suspected COVID-19 diagnosis was confirmed if SARS-CoV-2 nucleic acid was detected by reverse transcriptase-polymerase chain reaction (RT-PCR) in a clinical specimen. The diagnosis of SARS-CoV-2 infection was defined as one positive oro-nasopharyngeal swab or bronchoalveolar liquid performed in duplicate for SARS-CoV-2 E and S gene by a RT-PCR. Stratification of COVID-19 severity was based on World Health Organization (WHO) criteria [16].

### 2.3. COVID-19 Treatments 

The patients were treated with ad interim routinely used therapy (RUT) as suggested by the provisional guidelines of the Italian Society of Infectious and Tropical Diseases (SIMIT) and the Italian Medicine Agency (AIFA) [17]. In detail, we have dexamethasone (6 mg daily for 10 days) plus low molecular weight heparins (prophylactic/therapeutic dosage) + remdesivir ± eventual antibiotic treatment. All patients included in the study were supported by oxygen therapy delivered via invasive mechanical ventilation with or without ECMO. Patients receiving VV-ECMO underwent percutaneous veno-venous cannulation: a large venous drainage cannula was placed into the femoral vein up to the inferior vena cava (IVC) and the return cannula carrying oxygenated blood into the jugular vein to the superior vena cava (SVC). The ECMO circuit used in our center was the Gettinge CARDIOHELP^®^ device with Gettinge HLS Set Advanced 7.0^®^ ECMO circuits. Anticoagulation was achieved with unfractionated heparin that was adjusted to a target activated partial thromboplastin time of 40 to 55 s.

### 2.4. Sample Collection and Microbiological Analysis

Three sets of blood cultures were drawn in Bactalert FA Plus (bioMerieux, Marcy-l’Étoile, France) at admission from every critical COVID-19 patient transferred to the ICU from the isolation ward. Blood cultures were also retaken if the patient developed clinical signs of sepsis, had persistent fevers or deteriorated clinically, without other plausible explanations. After microscopic examination, fungal isolation was achieved by culturing positive blood cultures on Candida chromagar (CHROMID Candida, bioMérieux) and Sabouraud-CAF agar (Sabouraud Gentamicin Chloramphenicol 2, bioMérieux). Fungal identification was performed by a matrix-assisted laser desorption/ionization time-of-flight mass spectrometry (MALDI-TOF MS) system (Bruker Daltonik GmbH, Bremen, Germany). A broth microdilution system (Sensititre Yeast Oneplates, ThermoFisher Scientific, Waltham, Massachusetts, US) was used for testing fungal susceptibilities for amphotericin-B, azoles and echinocandins, following manufacturer’s instructions. The guidelines of the European Committee on Antimicrobial Susceptibility Testing (EUCAST) were used for the interpretations of fungal isolate susceptibility [18]. 

### 2.5. Clinical Definitions

Candidaemia episodes were diagnosed if *Candida* sp. was recovered from ≥1 blood culture, according to the ESCMID 2012 guidelines [19]. Sepsis was defined according to SEPSIS-3 criteria and ARDS was identified according to the 2012 Berlin criteria [20,21]. The “Candida score” was calculated by adding points provided by each component (severe sepsis, total parenteral nutrition, surgery, and multifocal Candida colonization). The total score was obtained by adding 1 point for each variable [22].

### 2.6. Statistical Analysis

Summary statistics are presented as frequencies and percentages for categorical data and median (interquartile range [IQR]) for continuous variables. Differences in categorical data were compared using the chi-square test or Fisher’s exact test, as appropriate. For continuous normally distributed two-group data, we compared differences using Student’s *t*-test or Mann–Whitney U if data were not normally distributed. The primary outcome was time to first candidaemia. We used competing risk modelling (Fine and Grey’s regression models) with time-on-study as the time scale to explore the effect of the exposure of interest (ECMO support) on the outcome incidence (candidaemia) considering the patient’s death as the competing event. Cumulative incidence functions (CIF) were plotted, and crude subdistribution hazard ratios (SHR) and their associated 95% confidence intervals (CIs) were calculated. To account for potential confounding, a propensity score (PS) based on the potential confounders of candidaemia and ECMO support was calculated using a logistic regression model. Then, quintiles of the resulting PS were added as a categorical covariate in the Fine and Grey models, using the lowest quintile as reference. Variables included in the PS model were: age (dichotomous using median age as cut-off: 0 ≤ 62 years; 1 ≥ 62 years); sex; ethnicity (=0 caucasian, 1 = non-caucasian); body mass index (continuous); chronic obstructive pulmonary disease (0 = no, 1 = yes); diabetes (0 = no, 1 = yes); hypertension (0 = no, 1 = yes), cardiovascular disease (0 = no, 1 = yes), ever smoking (0 = no, 1 = yes); SAPS II score (continuous); prone therapy (0 = no, 1 = yes); antiviral therapy (0 0 = no, 1 = yes); parental nutrition (0 = no, 1 = yes). Balance of variables used in calculating the PS was checked through standardized mean difference, using a cut-off of 0.20. The multivariable analysis was additionally adjusted for factors known at the time of potential exposure that could confound the association of interest to the outcome. To avoid a ratio events/predictors lower than 5, the Wald test was used to assess the contribution of adjusting variables to the model [23]. Adjusting variables included in the final model were: tocilizumab therapy (0 = no, 1 = yes); non-candidaemic BSI (0 = no, 1 = yes); candida score (continuous); inotropes therapy (0 = no, 1 = yes); vasopressor therapy (0 = no, 1 = yes). As a sensitive analysis, the PS was removed from the Fine and Gray model. The proportional hazards assumption was checked by testing the statistical significance of interaction terms involving failure time, each one at a time. Lastly, to assess predictors associated with candidaemia stratified on ECMO support, a logistic regression model was built for each group. Due to a limited number of events, variables included in the model were limited to the following: age (dichotomous using median age as cut-off: 0 ≤ 62 years; 1 ≥ 62 years); gender; candida score (continuous); non-candidaemic BSI (0 = no, 1 = yes). All statistical calculations were performed using Stata (StataCorp LLC, 4905 Lakeway Drive, College Station, TX, USA) version 17.0. A two-sided *p*-value < 0.05 was considered statistically significant.

### 2.7. Ethics Committee Approval

The Strengthening the Reporting of Observational Studies in Epidemiology (STROBE) guidelines were followed to create the study. The Ethics Committee of Policlinico Umberto I approved the study with number 109/2020.

## 3. Results

### 3.1. Characteristics of the Patients

During the period between March 2020 and June 2021, 138 patients were hospitalized in intensive care and underwent tracheal intubation and mechanical ventilation. Among these, 45 patients (32.6%) received ECMO support, while 93 patients (67.4%) suffering from severe hypoxemic acute respiratory failure did not meet ECMO criteria and were considered the control group. Overall, the median length of stay was 20 days (IQR 14–30), but patients with ECMO support had longer hospitalization than patients without support (27 days, IQR 17–38 vs. 18 days, IQR 13–27, *p*-value < 0.001). The median length of ECMO support for the 45 patients was of 18 days (IQR 12–23). Regarding the anamnestic characteristics and clinical severity at hospital admission, patients with ECMO support did not show statistically significant differences compared to unsupported ones, with the exception of age (51 years, IQR 40–58 vs. 69 years, IQR 60–76; *p*-value < 0.001), SAPS II score (24, IQR 24–27 vs. 34, IQR 28–42; *p*-value < 0.001), hypertension (29% vs. 48%; *p*-value 0.03) hepatopathy (9% vs. 0%; *p*-value 0.004), vasopressor therapy (58% vs. 86%, *p*-value < 0.001), inotropes therapy (7% vs. 38%, *p*-value < 0.001), bacterial colonization (31% vs. 48%, *p*-value 0.023), and BSI (53% vs. 34%, *p*-value 0.034). Finally, there was no statistically significant difference in mortality between the ECMO supported patients and unsupported ones (82% vs. 76%, *p*-value 0.43) (Table 1).

### 3.2. Impact of ECMO Support on Candidaemia

We observed an occurrence of 16 episodes of candidaemia among the 45 patients supported with ECMO, while only 13 of 93 unsupported patients were affected (36.0% vs. 14.0%, *p*-value 0.004), and there were no differences in the *Candida* species involved (Table 2).

At the survival analysis, patients in the ECMO group had a greater occurrence of candidemia than the patients of the unsupported group (SHR: 2.86, 95% CI: 1.39–5.88) (Figure 1). This finding was confirmed by the multivariable analyses with PS stratification, where patients in need of ECMO support showed higher sub-distribution hazard for candidemia (aSHR: 3.91, 95% CI: 1.73–8.86). As for the adjustment factors, having a higher candida score seemed to increase the hazard for candidemia occurrence (aSHR: 3.04, 95% CI: 2.09–4.42), while vasopressor therapy was negatively associated with the outcome (aSHR: 0.15, 95% CI: 0.05–0.43) (Table 3). The sensitive analysis confirmed a significant effect of ECMO support on the candidemia onset, although it was attenuated (aSHR 2.31, 95% CI: 1.15–4.64) (Appendix A). 

### 3.3. Stratified Analysis of Predictor for Candidaemia

The analysis of predictors associated with candidemia stratified by ECMO support showed that only the candida score had an impact on the odds of Candidemia occurrence in patients who needed ECMO support (aOR 27.73, 95% CI: 1.79–419.79). ECMO unsupported patients were influenced by the candida score, although to a lesser extent (aOR 2.14; 95% CI: 1.14–4.00), and by having been affected by at least one previous non-candidemic BSI (aOR 5.05, 95% CI: 1.23–20.67) (Appendix A).

## 4. Discussion

In this retrospective study, the incidence of candidemia was significantly higher in critically ill COVID-19 patients supported with VV-ECMO than in critically ill COVID patients who did not meet criteria for VV-ECMO (36.0% vs. 14.0%, *p*-value 0.004). Previous data on invasive candidiasis among severe COVID-19 patients ranged between 0.8 and 14% with an increased rate of candidemia, especially in comparison to the historical non-COVID-19 cohort [4,24]. A recent study reported an overall incidence of candidemia of 4.9% for ICU patients with severe COVID-19; however, the incidence increased to 36% in the subgroup of COVID-19 patients undergoing ECMO support [25]. The high incidence of candidemia observed in critically ill COVID-19 patients could be mainly explained by a mix of potential predisposing factors. In this study, patients ventilated for ≥10 days were included: the endogenous colonization of *Candida* species requires a period of 7–10 days for the development of IC after exposure to the risk factors [26].

Physiologically, the fungal microbiome can act as a reservoir for commensal fungi such as *Candida* species. Anyway, in immunocompromised hosts or under antibiotic treatment, they can act as pathogenic microorganisms and cause Candida infections. The COVID-19 patients are at an increased risk of candidemia and mycoses in case of concomitant conditions or pathologies that reduce immunity or favor fungal infections (i.e., diabetes, malignancy, multiple uses of steroids or antibiotics, chronic kidney diseases) [27]. In particular, it is well known that antibiotics used to treat bacterial superinfections can contribute to further gut microbiota perturbations and promote *Candida* overgrowth in SARSCoV2 infected patients [28,29]. Moreover, an overall gastrointestinal dysbiosis was reported as largely affecting SARSCoV2 infected patients and possibly related with the severity of the related disease [29]. In particular, SARSCoV2 infection induces gut microbiota dysbiosis with depletion of specific populations of commensal bacteria such as Bacteroidaceae, Lachnospiraceae and Ruminococcaceae [30,31]. Interestingly, gut dysbiosis was found to be associated with significant alterations in fecal mycobiomes of SARSCoV2 infected subjects. In fact, compared to healthy controls, the fungal burden was found higher in COVID-19 patients and mainly characterized by enrichment of Candida spp in hospitalized patients [32,33]. Finally, a transient immunodeficiency that potentially exposes patients to opportunistic infections were described associated with SARSCoV2 infection also in immunocompetent individuals [34]. Particularly, an increased susceptibility for *Candida* infections in critically ill COVID-19 patients was suggested by the observation of impaired immune response to those pathogens characterized by the abrogated release of IL-6, TNF, IL-1α, and IL-1β toward *Candida albicans* [35]. 

Regarding the etiology, a recent review of the studies on the topic showed an upsurge in the cases due to non-albicans *Candida* species despite the fact that *C. albicans* remains the most common cause of Candidemia observed among the COVID-19 patients [25]. In our study, we observed a higher incidence of infections due to *C. parapsilosis* than to *C. albicans* and *C. tropicalis*, with no difference between ECMO supported and unsupported patients. Probably, the increased incidence of *C. parapsilosis* (of which the tropism for inert materials is well-known) was linked with the specific characteristic of our cohort composed by critically ill patients with severe COVID-19 prognosis and heavily treated with the high invasive ICU procedures and/or ECMO (with prolonged exposure to a significant volume of endovascular foreign materials) [36]. 

The analysis of predictors associated with candidemia in our study showed that only the candida score had an impact on the odds of Candidemia occurrence in patients who needed ECMO support (aOR 27.73, 95% CI: 1.79–419.79). On the other hand, the candida score (aOR 2.14; 95% CI: 1.14–4.00) and a previous episode of non-candidemic BSI (aOR 5.05, 95% CI: 1.23–20.67) were predictors associated with candidemia in ECMO unsupported patients. Interestingly, the candida score ranged between 1 and 4 in patients with candidemia enrolled in our study. The observation indicates that this scoring tool, while remaining a good predictor of candidemia, tends to underestimate the risk in the setting of severe COVID-19. In fact, invasive candidiasis is considered highly improbable in subjects with candida score < 3, and rates of invasive candidiasis are estimated at 0% in patients with score = 2 or 3, and 17.6% in patients with score = 4 in ICU patients [37]. Moreover, other studies showed that use of central lines, mechanical ventilation, biologic, steroid and paralytics drugs were found to be independent risk factors for candidemia in severe COVID-19 patients [38,39]. These differences may be due to sample selection; in fact, our cohort was completely composed of ICU critically ill patients and an intensive care unit-level care was a characteristic common to all patients; likewise, immunomodulant drugs were wildly used.

The use of ECMO assistance was found related with the risk of candidemia in this study: in particular, the potential risk is linked to the long duration of ECMO support which can lead to intestinal ischemic complications and thus favor microbial translocation [40]. About the clinical outcomes of the candidemia in SARSCoV2 infected patients, recent studies reported an overall increase in the mortality [27]. In our cohort, there was no statistically significant difference in mortality between the ECMO supported patients and unsupported ones. Both groups were extremely severe patients: patients in the non-ECMO group had high SAPS II scores upon admission to the ICU. It is interesting to note that these were significantly more hypertensive patients with likely lower age-related vascular reactivity, also confirmed by a greater need for vasopressors and isotropic drugs [41]. One would presume that older patients and those who were more critically ill would have a higher incidence of candidemia. On the other hand, patients in the ECMO group had longer ICU stays because they were initially less severe and candidates for ECMO support, but often received bridging therapy without improvement.

To the best of our knowledge, this is the first study focused on the incidence of fungal infection in critically ill COVID-19 patients undergoing ECMO support. It is of paramount importance to highlight the possible risk factors in this special population. However, this study has several limitations: first, it is a retrospective study, but the control group has been powered using a propensity score technique, a prospective study should be designed to better assess the impact of different risk factors. The use of renal replacement therapy is also associated with the occurrence of candidemia, but we could not investigate this potential risk factor in the propensity score model because only a few patients in both groups were treated with this technique [25]. Second, the role of antifungal therapy was not analysed in this study. Third, local fungal epidemiology may have affected candidemia rates in both ECMO and non-ECMO patients, but the particular impairment of the immune system may play a role on this result. Finally, the study did not aim to specifically determine any ‘‘break points’’ in ECMO duration as related to risk of fungal infection, but such analysis may prove to be beneficial in the future for quantifying and controlling for all the variables identified in COVID-19 critically ill patients undergoing ECMO.

## 5. Conclusions

In this retrospective cohort study, the incidence of candidemia was significantly higher in critically ill COVID-19 patients supported by VV-ECMO than in non ECMO patients caused by the impaired immune response. Further studies are needed to better understand the risk factors and mechanisms underlying this marked incidence.

## Figures and Tables

**Figure 1 jof-09-00119-f001:**
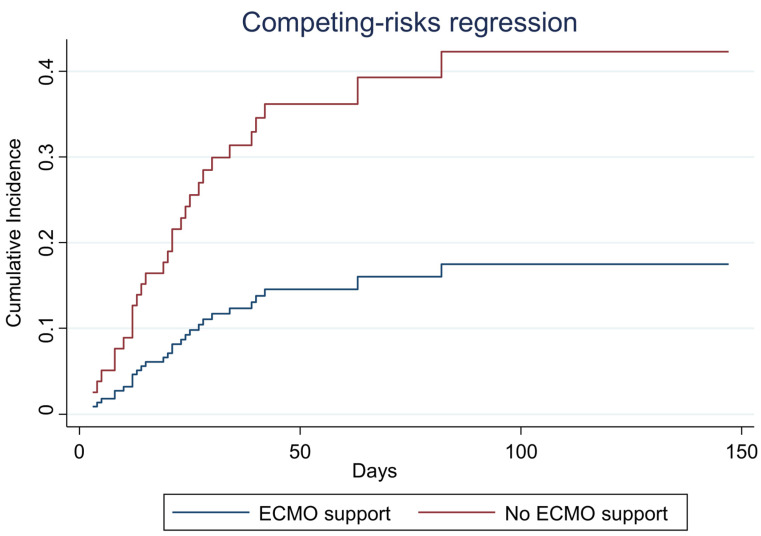
Cumulative incidence function for candidaemia among the patients admitted to the intensive care unit of Umberto I teaching hospital of Rome between March 2020 and June 2021 and who underwent IOT and mechanical ventilation.

**Table 1 jof-09-00119-t001:** Characteristics of patients admitted to the intensive care unit of Umberto I teaching hospital of Rome between March 2020 and June 2021 and who underwent IOT and mechanical ventilation. Results are expressed as number (percentage) or median (interquartile range).

	ECMO Support		
Patients	Yes	No	*p*-Value
N	45	93	
Age, years	51 (40–58)	69 (60–76))	<0.001
Gender			
Male	34 (75.6)	63 (67.7)	0.35
Female	11 (24.4)	30 (32.3)	
BMI	29 (25–34)	28 (26–31)	0.17
Pre-existing comorbidity			
Chronic obstructive pulmonary disease	3 (6.7)	14 (15.1)	0.16
Diabetes	8 (17.8)	18 (19.4)	0.82
Renal Insufficiency	0 (0)	2 (2.2)	0.32
Hypertension	13 (28.9)	45 (48.4)	0.03
Cardiovascular Disease	6 (13.3)	15 (16.1)	0.67
Chronic liver failure	4 (8.9)	0 (0.0)	0.004
Ever Smoker	3 (6.7)	7 (7.5)	0.86
Ethnicity	9 (20.0)	8 (8.6)	0.056
SAPS II	24 (24–27)	34 (28–42)	<0.001
Medical therapy	45 (100)	92 (98.9)	0.49
Antibiotic therapy	45 (100)	92 (98.9)	0.49
Tocilizumab	5 (11.1)	22 (23.7)	0.082
Antiviral Drugs	23 (51.1)	54 (58.1)	0.44
Parenteral nutrition	22 (48.9)	32 (34.4)	0.1
Inotropes	3 (6.7)	35 (37.6)	<0.001
Vasopressor	26 (57.8)	80 (86.)	<0.001
Prone Therapy	17 (37.8)	35 (37.6)	0.99
ICU Stay, days	27 (17–38)	18 (13–27)	<0.001
Lung bacterial superinfection	27 (60.0)	52 (55.9)	0.65
Bacterial colonization	31 (68.9)	45 (48.4)	0.023
BSI	24 (53.3)	32 (34.4)	0.034
Non-candidaemic BSI	19 (42.2)	29 (31.2)	0.2
Candida score	1 (1–3)	1 (0–2)	0.78
Mortality rate	37 (82.2)	71 (76.3)	0.43

ECMO: Extracorporeal Membrane Oxygenation; SAPS: Simplified Acute Physiology Score; ICU; Intensive Care Unit; BSI: Blood Stream Infection.

**Table 2 jof-09-00119-t002:** Candidaemia cases and *Candida* species involved in patients admitted to the intensive care unit of Umberto I teaching hospital of Rome between March 2020 and June 2021 and who underwent IOT and mechanical ventilation. Results are expressed as number (percentage).

	ECMO Support		
Patients	Yes	No	*p*-Value
N	45	93	
Candidaemia	16 (36.0)	13 (14.0)	0.004
*Candida* species			
*albicans/tropicalis*	5 (31.3)	4 (30.8)	0.98
*Parapsilosis*	11 (68.8)	9 (69.2)	

**Table 3 jof-09-00119-t003:** Multivariable competing risk Fine-Gray regression models for candidaemia (among the patients admitted to the intensive care unit of Umberto I teaching hospital of Rome between March 2020 and June 2021 and who underwent IOT and mechanical ventilation).

	Candidaemia	
	aSHR (95% CI)	*p*-Value
ECMO support	3.91 (1.73–8.86)	0.001
Non-candidaemic BSI	1.61 (0.65–4.01)	0.308
Tocilizumab therapy	0.40 (0.12–1.40)	0.153
Inotropes Therapy	0.97 (0.29–3.18)	0.957
Vasopressor Therapy	0.15 (0.05–0.43)	<0.001
Candida score	3.04 (2.09–4.42)	<0.001
Propensity score quantiles	4.15 (1.78–9.64)	0.001
I quintile	Ref.	
II quintile	1.96 (0.43–8.99)	0.387
III quintile	0.68 (0.17–2.80)	0.593
IV quintile	0.25 (0.06–1.09)	0.064
V quintile	0.52 (0.11–2.44)	0.408

aSHR: Adjusted Subdistribution Hazard Ratio; CI: Confidence Interval. ECMO: Extracorporeal Membrane Oxygenation; BSI: Blood Stream Infection; Ref; reference.

## Data Availability

Data about investigated patients and analytic methods will not be made publicly available.

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
