# Peer review of "High Incidence of Candidemia in Critically Ill COVID-19 Patients Supported by Veno-Venous Extracorporeal Membrane Oxygenation: A Retrospective Study"

_jof, 2023, doi:10.3390/jof9010119_

Round 1
Reviewer 1 Report
Dear authors,
This is an interesting manuscript about the incidence of candidemia in severe Covid-19 patients. I have some comments and recommendations.
Abstract - line 17 - please define BSI
Introduction
-line 44 - define CRRT
Materials and Methods
-comparing the gross weights of the two groups is not relevant, especially since the BMI comparison shows that there are no statistically significant differences.
Results
-how do you explain the significant difference in the presence of hypertension (more in the non ECMO group)?
Discussion
-please reduce the rate of self citation (Ref No 28-31).
Author Response
Q1- Abstract - line 17 - please define BSI
R1- We agree with your observation about the abstract and we have defined BSI. See Page 1 line 17
Q2- line 44 - define CRRT
R2- Thank you, we have corrected: see Page 2 line 45
Q3- comparing the gross weights of the two groups is not relevant, especially since the BMI comparison shows that there are no statistically significant differences.
R3- Thank you for this comment, we have removed the gross weights from table 1
Q4- how do you explain the significant difference in the presence of hypertension (more in the non ECMO group)?
R4- We are grateful to the reviewer for this important observation. Hypertension is a common comorbidity in COVID-19 patients. In addition, its incidence increases in the elderly. In this study, patients in the non-ECMO group were older than those in the ECMO group. We preferred not to include this comment in the text, however we highlight the data in the results.
Q5- please reduce the rate of self citation (Ref No 28-31).
R5- We agree and we have reduced the rate of self-citation (removed n°28 and n°31). See references

Reviewer 2 Report
In this single-center retrospective study in Italy, the study authors attempted to analyze the incidence of COVID-19 VV-ECMO patients with Candidemia, their mortality rate and their time to in-hospital all-cause mortality. They noted a significantly higher incidence of Candidemia in VV-ECMO patients, with the higher Candida score increasing the risk of Candidemia occurrence.
Overall, the study is comprehensive and well designed, with appropriate risk factors for Candidemia being analyzed. The finding of significantly higher Candidemia in ECMO patients in the study should raise suspicion in clinicians taking care of these critically ill patients, with the adjunctive use of the Candida score to help in their assessment. I had a few points below to help strengthen the manuscript’s impact.
Major Comments
1. The study authors note that they included patients who had been on ventilator support for ≥10 days (page 2 of 11, line 60). Was there a particular reason for including this time cut-off? I understand that the data regarding timing of candidemia varies from study to study and that this may have been selected arbitrarily. It would be helpful if clarification was provided in this regard.
2. Use of renal replacement therapies have also been noted to be associated with the occurrence of Candidemia (including in this recent study by Blaize et al doi:10.3390/jof8070678). Did the study authors investigate it as a potential risk factor in their propensity score model? Or were there only a smaller number of patients who needed these therapies and it was thought not to be a confounding factor? It would be helpful if the study authors evaluate and include this, or consider commenting on it.
3. It is interesting to note that the ECMO patients were significantly younger and had significantly lower proportions of need for vasopressor therapy and inotrope therapy (Page 4 of 11, lines 172-175). One would presume that older patients and those who were more critically ill would have a higher incidence of Candidemia. Did the study authors investigate this further? It would be helpful if they highlight these in the Discussion.
4. In the limitations, the study authors should also consider adding that the local fungal epidemiology may have impacted their Candidemia rates in both the ECMO and non-ECMO patients, particularly given the widely varying rates reported in the literature (as described in Page 7 of 11, Lines 219-224).
Minor Comments
1. Page 1 of 11, line 32 – the full-form of the abbreviation of SARS-CoV-2 should be utilized here, as it is the first instance in the manuscript of the abbreviation.
2. Page 1 of 11, line 35 – the full-form of the abbreviation of COVID-19 should be utilized here, as it is the first instance in the manuscript of the abbreviation.
3. Page 2 of 11, line 72 – the study authors mention use of “anamnestic data”. I am not particularly familiar with this term. Does it refer to serologic data? Please clarify this.
4. When describing the impact of the Candida score in the patients who needed ECMO support (page 5 of 11, line 196-197 and page 8 of 11, line 260-261), the study authors mention “the candida score had an impact on the odds of patients who needed ECMO support (aOR 260 27.73, 95% CI: 1.79-419.79)”. For sake of clarity, they could consider modifying the line as “the candida score had an impact on the odds of Candidemia occurrence in patients who needed ECMO support”.
5. The study authors have used Candida score at many times in the manuscript. For readers who may not be familiar with the scoring system, it would be helpful to include a brief summary and reference [León et al doi:10.1097/01.CCM.0000202208.37364.7D] in the Definitions.
Author Response
Major Comments
Q1- The study authors note that they included patients who had been on ventilator support for ≥10 days (page 2 of 11, line 60). Was there a particular reason for including this time cut-off? I understand that the data regarding timing of candidemia varies from study to study and that this may have been selected arbitrarily. It would be helpful if clarification was provided in this regard.
R2- Thank you for this important comment. One major cause of severe candidiasis is the endogenous colonization of Candida species, which requires a period of 7-10 days for the development of IC after exposure to the risk factors (Eggimann P et al. doi.org/10.1016/j.jhin.2014.11.006). In addition, the median time for obtaining positive blood cultures is 2–3 days (possibly up to ≥7 days) (Pappas PG et al. DOI: 10.1093/cid/civ933). Thus, for a patient with the confirmed diagnosis of candidaemia at 8 days after ICU admission, the endogenous colonization of Candida species might have actually occurred on or before the first day of ICU admission. Similarly, for a patient with the confirmed diagnosis of candidaemia at 12-13 days after ICU admission, the endogenous colonization of Candida species might have occurred 3–5 days after ICU admission. This cut-off is now discussed in the text see page 7 line 231-233
Q2- Use of renal replacement therapies have also been noted to be associated with the occurrence of Candidemia (including in this recent study by Blaize et al doi:10.3390/jof8070678). Did the study authors investigate it as a potential risk factor in their propensity score model? Or were there only a smaller number of patients who needed these therapies and it was thought not to be a confounding factor? It would be helpful if the study authors evaluate and include this, or consider commenting on it.
R2- We agree with the reviewer about this point; however only few patients of both groups were treated with CRRT and we pointed out this data as a limit. Page 9 Line 302-305
Q3- It is interesting to note that the ECMO patients were significantly younger and had significantly lower proportions of need for vasopressor therapy and inotrope therapy (Page 4 of 11, lines 172-175). One would presume that older patients and those who were more critically ill would have a higher incidence of Candidemia. Did the study authors investigate this further? It would be helpful if they highlight these in the Discussion.
R3- Thank you for this comment. Patients in the non-ECMO group were older, more hypertensive, and a higher percentage of them required vasopressors and inotropic drugs. All of these factors seem to suggest an increased risk of candidemia related to a worse condition, however, patients in the ECMO group who met the inclusion criteria for extracorporeal bridge treatment had a prolonged length of stay in the ICU and were at higher risk of developing candidemia. See Page 8 Line 290-294.
Q4- In the limitations, the study authors should also consider adding that the local fungal epidemiology may have impacted their Candidemia rates in both the ECMO and non-ECMO patients, particularly given the widely varying rates reported in the literature (as described in Page 7 of 11, Lines 219-224).
R4- This is an important point to highlight; we have implemented the limitations as you suggested. See Page 9 Line 306-308.
Minor Comments
Q1- Page 1 of 11, line 32 – the full-form of the abbreviation of SARS-CoV-2 should be utilized here, as it is the first instance in the manuscript of the abbreviation.
R1- Thank you for the comment, the text has been corrected. See page 1 line 32
Q2- Page 1 of 11, line 35 – the full-form of the abbreviation of COVID-19 should be utilized here, as it is the first instance in the manuscript of the abbreviation.
R2- the text has been corrected. See page 1 line 35
Q3- Page 2 of 11, line 72 – the study authors mention use of “anamnestic data”. I am not particularly familiar with this term. Does it refer to serologic data? Please clarify this.
R3- thank you, we clarified this point. See page 2, line 74-76
Q4- When describing the impact of the Candida score in the patients who needed ECMO support (page 5 of 11, line 196-197 and page 8 of 11, line 260-261), the study authors mention “the candida score had an impact on the odds of patients who needed ECMO support (aOR 260 27.73, 95% CI: 1.79-419.79)”. For sake of clarity, they could consider modifying the line as “the candida score had an impact on the odds of Candidemia occurrence in patients who needed ECMO support”.
R4- Thank you for the clarification, we have corrected. See page 5, line 196-197 and page 8, line 267-268
Q5- The study authors have used Candida score at many times in the manuscript. For readers who may not be familiar with the scoring system, it would be helpful to include a brief summary and reference [León et al doi:10.1097/01.CCM.0000202208.37364.7D] in the Definitions.
R5- We include the Candida score definition in the methods see page 3 line 120-122

Reviewer 3 Report
The authors present the results of an epidemiological study showing a positive correlation of ECMO ventilation support and candidemia in hospitalized COVID-19 patients. The manuscript is well-written and describes a thorough statistical analysis of patient data. The statistical methods using propensity scores are sound and account for the possible confounding effects of the retrospective, non-randomized study design. The discussion is appropriately limited to the interpretation of the presented data, does not speculate about mechanisms, and accurately points out the limitations of the study. With its solid execution and identification of risk factors for candidemia in the COVID-19 patient population, the study makes a important contributions to the body of knowledge of nosocomial fungal infections and merits publication in JoF.
Author Response
Q1- The authors present the results of an epidemiological study showing a positive correlation of ECMO ventilation support and candidemia in hospitalized COVID-19 patients. The manuscript is well-written and describes a thorough statistical analysis of patient data. The statistical methods using propensity scores are sound and account for the possible confounding effects of the retrospective, non-randomized study design. The discussion is appropriately limited to the interpretation of the presented data, does not speculate about mechanisms, and accurately points out the limitations of the study. With its solid execution and identification of risk factors for candidemia in the COVID-19 patient population, the study makes a important contributions to the body of knowledge of nosocomial fungal infections and merits publication in JoF.
R1- we are grateful to you for your particularly attentive reading of our work. Your very detailed summary of the paper does not require any particular response.

Round 2
Reviewer 1 Report
Dear authors,
I think your manuscript is improved and the study worth to be published.
Thank you.